# Functional Genomics Analysis to Disentangle the Role of Genetic Variants in Major Depression

**DOI:** 10.3390/genes13071259

**Published:** 2022-07-15

**Authors:** Judith Pérez-Granado, Janet Piñero, Alejandra Medina-Rivera, Laura I. Furlong

**Affiliations:** 1Research Programme on Biomedical Informatics (GRIB), Hospital del Mar Medical Research Institute (IMIM), Department of Medicine and Life Sciences (MELIS), Universitat Pompeu Fabra (UPF), Dr. Aiguader 88, 08003 Barcelona, Spain; jperez2@imim.es (J.P.-G.); janet.pinero@upf.edu (J.P.); 2MedBioinformatics Solutions SL, Almogàvers 165, 08018 Barcelona, Spain; 3Laboratorio Internacional de Investigación sobre el Genoma Humano, Universidad Nacional Autónoma de México, Campus Juriquilla, Blvd Juriquilla 3001, Santiago de Querétaro 76230, Mexico

**Keywords:** major depression, genetic variants, eQTL, colocalization analysis, transcription factors, genetic regulation

## Abstract

Understanding the molecular basis of major depression is critical for identifying new potential biomarkers and drug targets to alleviate its burden on society. Leveraging available GWAS data and functional genomic tools to assess regulatory variation could help explain the role of major depression-associated genetic variants in disease pathogenesis. We have conducted a fine-mapping analysis of genetic variants associated with major depression and applied a pipeline focused on gene expression regulation by using two complementary approaches: cis-eQTL colocalization analysis and alteration of transcription factor binding sites. The fine-mapping process uncovered putative causally associated variants whose proximal genes were linked with major depression pathophysiology. Four colocalizing genetic variants altered the expression of five genes, highlighting the role of SLC12A5 in neuronal chlorine homeostasis and MYRF in nervous system myelination and oligodendrocyte differentiation. The transcription factor binding analysis revealed the potential role of rs62259947 in modulating P4HTM expression by altering the YY1 binding site, altogether regulating hypoxia response. Overall, our pipeline could prioritize putative causal genetic variants in major depression. More importantly, it can be applied when only index genetic variants are available. Finally, the presented approach enabled the proposal of mechanistic hypotheses of these genetic variants and their role in disease pathogenesis.

## 1. Introduction

Major Depression (MD) is the leading cause of impairment around the world [1]. It is mainly treated with both psychotherapy and drugs, but the latter is only effective in 40% of the patients [2]. Currently, there are no available biomarkers or tests that can aid in either MD diagnosis or personalized treatment. As a complex disease, multiple genetic variants (GVs) have been associated with MD in Genome-Wide Association Studies (GWAS), most of them falling within non-coding regions of the genome [3,4].

Functional follow-up studies to unravel the regulatory mechanisms by which these GVs play a role in the disease are key to understanding the molecular underpinnings of the disease and identifying biomarkers or new drug targets. Some authors propose that the efforts should be centered on the interpretation of GWAS signals to identify the causal GVs, meaning those with a biological effect on a disease, and their regulatory potential, instead of pursuing more GWAS [5].

In this study, we have focused on the GWAS meta-analysis on MD performed in 2019 by Howard et al. [3]. Full-genome summary statistics are not publicly available for this GWAS, so we have leveraged available data on index GVs. Ninety-seven loci were identified as significantly associated with MD, and these underwent the classic post-GWAS analysis: a gene-set enrichment analysis, the computation of polygenic risk score, and genetic correlation with other traits, as well as drug-gene interaction analysis. In line with previous GWAS findings, most GVs lie in non-coding regions, thus having no obvious direct effect on a gene.

A necessary step forward to disentangle the role of GVs identified in GWAS requires the evaluation of functional regulatory variation. Here, we have pursued two complementary analytical approaches geared toward the use of index GVs: (1) identification of candidate susceptibility genes using expression quantitative trait loci in cis (cis-eQTLs), which are enriched among disease-associated loci [6], and (2) characterization of transcription factor (TF) binding sites modified by GVs, which are key to understanding their potential impact on regulatory mechanisms [6,7,8].

In the present study, we aim to advance the understanding of MD molecular underpinnings. We have designed and applied a regulatory variation analysis pipeline and conducted a functional enrichment analysis of the GVs, either acting as eQTLs or altering the transcription factor binding site (TFBS), along with the proximal (pGenes) and regulated genes (eGenes). Our findings provide biological insights into the functional role of MD GVs and enable the proposal of mechanistic hypotheses. 

## 2. Materials and Methods

### 2.1. MD GWAS Dataset and LD expansion

In order to obtain a comprehensive and reliable set of genetic variants (GVs) associated with major depression (MD), we focused our analysis on the GWAS meta-analysis from Howard et al. [3]. This meta-analysis evaluated 807,553 European individuals (246,363 cases and 561,190 controls) and identified 102 genetic variants (GVs) associated with MD. We retrieved these data from the summary statistics available at GWAS Catalog [9] (Accession Study: GCST007342, note that the full-genome summary statistics for this GWAS were not publicly available; downloaded in December 2020). We filtered the GVs by genome-wide significance (*p*-value ≤ 5 · 10^−8^ and proceeded with the analysis with this set. We then fine-mapped MD-associated GVs to prioritize the causal ones using the Probabilistic Identification of Causal SNPs (PICS) algorithm [10]. In brief, PICS takes the most significant variant per association locus and performs LD expansion using the 1000 Genomes Project linkage disequilibrium (LD) information data for the study population and then identifies the GVs more likely to be causal (PICS probabilities). Using the PICS2 Data portal, we downloaded the precomputed PICS GVs for this study. This data constituted our full dataset of GVs. 

### 2.2. GVs Annotation: VEP, CADD and ENCODE

We annotated the full set of GVs with Variant Effect Predictor (VEP) [11] and Combined Annotation Dependent Depletion (CADD) [12]. VEP annotates GVs’ consequence type using the Sequence Ontology, its allele frequency from the 1000 Genomes Project Phase 3 along with the genomic coordinates, chromosome, and mapped gene at ±5000 bp distance (from now on pGenes). Combined Annotation Dependent Depletion (CADD) assesses GVs’ potential pathogenicity by evaluating the PHRED-like scaled C-score; the recommended cut-off ≥ 15 was set to identify potentially pathogenic variants.

We analyzed the GVs with the Encyclopedia of DNA Elements (ENCODE) [13] to identify those potentially lying in transcription factor binding sites (TFBS). ENCODE data analysis was performed using SNPNexus [14], an online platform that allows a comprehensive annotation of GVs by integrating multiple tools.

### 2.3. Fine-Mapping and Colocalization of GWAS and cis-eQTLs

PICS2, in addition to GWAS PICS GVs, has precomputed PICS GVs for all Genotype-Tissue Expression (GTEx) V8 best eQTLs per gene, per tissue type. We overlapped the extracted GWAS PICS for MD GVs with GTEx cis-eQTL PICS GVs, filtering both sets by a PICS probability greater than 10% to narrow down the set to the most likely causal GVs without being overly permissive, as previous applications of this method have done [15]. We performed a Fisher test to assess the enrichment of GVs in eQTL regions. Finally, to identify colocalizing GWAS and eQTL GVs, we computed the products of PICS probabilities following the colocalization posterior probability (CLPP) method, which assumes independence of causal probabilities for GWAS and eQTL GVs [16]. The genes regulated by these eQTLs from now on will be referred to as eGenes.

### 2.4. TF Binding Analysis with RSAT Variation Tools

We predicted those GVs affecting the TFBS using the Regulatory Sequence Analysis Tools (RSAT) suite, which evaluates cis-regulatory elements. First, we used ENCODE ChIP-seq data to keep only the GVs lying in TFBS and, therefore, have a more biologically relevant set of GVs and reduce the number of tests. However, ChIP-seq data retrieve regions of around 100–1000 bp, but the actual binding site corresponds to 9–15 bp [17,18]. Thus, we proceeded with the RSAT analysis for a more robust and accurate assessment of the GVs potentially altering the TFBS. RSAT provides tools that evaluate cis-regulatory elements to predict GVs affecting the TFBS by modifying the transcription factor (TF) binding affinity. 

RSAT modular structure allowed the concatenation and independent execution of programs, each with a different goal. Before scanning the GVs and in order to account for their different nucleotide composition, we created four sets of background models according to the GV’s functional impact obtained with VEP (i.e., intergenic and UTR, intronic, regulatory, and non-coding GVs). The subsequent steps were performed for each set separately. The module *create-background-model* was executed using the sequences obtained with *fetch-sequences-from-UCSC*, with the peak regions retrieved by ENCODE as input. In parallel, the module *retrieve-variation-sequence* was used to obtain the flanking sequence (30 bp per side) of the GVs of interest, using the dbSNP, genomic coordinates, reference, and alternative allele.

To assess the TFBS alterations, position weight matrices (PSSM) for TFs expressed in brain tissues (filtering them using GTEx expression data, ≥2 transcripts per million (TPM)) [19] were retrieved from the following databases: JASPAR [20], ENCODE, HOCOMOCO [21], footprintDB [22], and hPDI [23]. In all cases, the non-redundant Homo sapiens database version was used.

Finally, the module *variation-scan* was run with the previously built background Markov models (order 2 to account for CpGs without overfitting), the PSSM matrices, the GVs with their flanking sequences (see above), and the following parameters: weight of predicted sites (>1), weight difference between variants (>1), *p*-value of predicted sites (<1 e-3), and *p*-value ratio between variants (>10). The weight represents the binding affinity and the *p*-value of a score is the probability of observing a score of at least weight given a background model.

In addition, two control datasets, one randomizing TF motifs and one randomizing GVs, were built to validate the results obtained running RSAT with the GVs of interest. On the one hand, the TF’s PSSMs matrices were permuted using *permute-matrix -perm 5* to get randomized matrices with the same nucleotide composition and information content. On the other hand, a control set of GVs (1:10) was built using vSampler [24] with the following parameters: distance to closest transcription start site (TSS) deviation (±5000), gene density deviation (±5 in 100 kbp), number of variants in LD (±50 and r^2^ = 0.1), controlling for coding/non-coding match and variant type specificity, excluding for input GVs and sampling across the chromosome. Both controls were analyzed with the described RSAT pipeline.

We compared our set of GV-TF motif pair *p*-value ratios against the distribution of *p*-value ratios for the given motif in both control datasets. A Wilcox test was used to evaluate the results obtained from the controls because normality of *p*-value ratio distribution could not be assumed for most motifs after running a Shapiro–Wilk test. The alternative hypothesis tested was “greater”.

In addition, to further confirm statistically significant GVs, a larger negative control dataset of GVs (1:1000) was generated. Again, vSampler was used with relaxed parameters to get a bigger control set (i.e., controlling for coding/non-coding match and variant type specificity, excluding for input SNPs, and sampling across chromosomes). The same non-parametric test was used to evaluate the results.

### 2.5. Identification of TF Active Regions with ChromHMM

We used chromatin state annotations from ChromHMM [25,26], available from ENCODE (v3), to evaluate whether GVs significantly altering the TFBS were lying in active transcription sites of brain regions. Under a 18-state ChromHMM model, we consider the following states annotations as active regulatory regions [26]: TssA, TssFlnk, TssFlnkU, TssFlnkD, Tx, TxWk, EnhG1, EnhG2, EnhA1, EnhA2, EnhWk, ZNF/Rpts. The available brain regions and cell types were: Brain Angular Gyrus, Brain Inferior Temporal Lobe, Brain Cingulate Gyrus, Brain Anterior Caudate, Brain Substantia Nigra, Brain Dorsolateral Prefrontal Cortex, Brain Hippocampus Middle, and Astrocytes. Additionally, the resulting TFs whose binding was altered were filtered by their expression in the specific brain region using GTEx matched data when available; otherwise, data for all brain regions were considered.

### 2.6. Retrieval of Regulation Evidence

We looked for evidence of gene expression regulation of TFs by matching GVs-TFs pairs from the TF binding analysis using RSAT with eQTL PICS GVs. We further explored the hTFtarget database [27] to identify specific mechanistic regulation evidence of those TFs whose binding is altered by our set of GVs to regulate the expression of the target eGenes. The hTFtarget database contains associations of TFs and their targets from chromatin immunoprecipitation sequencing (ChIP-seq) in a specific tissue. We considered evidences for mechanistic regulation when eQTL and ChIP-seq data tissues matched. 

### 2.7. pGenes, eGenes, and GVs Characterization

We conducted a gene-set enrichment analysis using the tool g:Profiler via the R package gprofiler2 [28], which integrates different resources and annotates enriched terms at the following levels: (1) biological processes, molecular functions, and cellular processes annotated with the Gene Ontology (GO); (2) pathways from Reactome (REAC) and WikiPathways; (3) miRNA annotations from MIRNA; (4) phenotypic features associated to disease from Human Phenotype, which is mainly focused on rare Mendelian disorders. In addition, we included DISGENET plus [29,30] association data (v16) in this analysis to evaluate the annotation of complex diseases and phenotypic traits; note that the study by Howard et al. was removed from this dataset to avoid circularity. Variant-set functional enrichment analysis was performed using variant association data from DISGENET Plus. We considered the set of known genes as the domain scope for the analysis. Furthermore, we characterized tissue expression using GTEx gene expression data (v8).

We performed these analyses for the following two gene-sets: (1) genes mapped to by MD-associated GVs (pGenes) and (2) genes regulated by cis-eQTLs (eGenes), and two variant-sets: (1) causal GVs and (2) colocalizing GVs.

## 3. Results

### 3.1. Major Depression Associated Genetic Variants Lie in Non-Coding Regions of the Genome

The GWAS study by Howard et al., 2019, reported 102 risk loci associated with major depression (MD), 97 with a *p*-value ≤ 5 · 10^−8^, which were the starting point of our analysis. After LD expansion, we obtained a set of 5723 potentially causal genetic variants (GVs) (Appendix A). We annotated these GVs with VEP [11] and CADD [12] (Appendix A). The identification of probable causal GVs using PICS fine-mapping GWAS data [10] revealed 172 GVs (PICS >10%) in LD with the 97 GWAS risk loci (Appendix A). These GVs are located in different regions of the genome, but most of them are in non-coding regions, being mainly annotated as intronic (30%), intergenic (30%), or located in non-coding transcript regions (17%) (Figure 1A). Only two GVs lie in exonic regions (i.e., synonymous and nonsynonymous consequence types). The median allele frequency of these GVs was 0.364 (with more deleterious consequence types having lower allele frequencies) (Figure 1B). Only 4% (7) of the GVs were predicted by CADD as potentially pathogenic (Figure 1C). The fine-mapped GVs were assigned to 95 proximal genes (±5000 bps), from now on referred to as pGenes. pGenes were classified based on their expression across tissues based on GTEx gene expression data [19]. Using hierarchical clustering, genes were divided into three roughly equally distributed clusters that seem to correspond to constitutively, lowly expressed in all tissues, and highly expressed in brain tissues (Appendix A). 

The pGenes are functionally enriched in GO terms related to nervous system development, neuron differentiation, synaptic signaling, and different cellular components of the neuron such as dendrite, axon, or synapse (Appendix A); these biological processes and molecular functions are involved in the pathophysiology of MD [31]. pGenes are associated with an abnormal nervous system morphology and physiology according to the Human Phenotype ontology. Disease enrichment analysis shows enrichment for the association of both pGenes and causal GVs with major depressive disorder and other related mental disorders such as schizophrenia or bipolar disorder (Figure 2 and Appendix A). pGenes are also associated with comorbid phenotypes and conditions in MD, such as smoking behavior, body mass index, and duration of sleep [32]. Notably, 37% of pGenes and 42% of GVs have no previous evidence of association with depression or other mental disorders.

Some of the pGenes are associated with processes related to MD pathogenesis, such as TLR4, involved in immune response [33], ESR2, a regulator of estrogen response [34], TCF4, with a role in nervous system development [35], DCC, in charge of axon guidance and neuronal connectivity [36], PAX5, which interferes in mouse neural stem cells proliferation and migration [37,38], and CYP7B1, that participates in the metabolism of the neurosteroids DHEA and pregnenolone [39]. Among the potentially pathogenic GVs, according to CADD, there are 3 intronic GVs lying in ZNF536, a gene involved in the negative regulation of neuron differentiation [40], a relevant process in MD pathogenesis and treatment [41]. rs1021362 lies in SORCS3, a gene previously associated with stress response associated with MD [37,42], rs3793577 lies in ELAVL2, whose silencing in animal models is associated with reduced behavioral despair [43]; the remaining GVs have been previously associated with major depression by several PheWAS studies [15]. 

### 3.2. Major Depression Causal Genetic Variants Regulate the Expression of Genes in Cis

The 172 fine-mapped GWAS GVs overlap with 13 GTEx PICS GVs (Figure 1), revealing an enrichment of MD causal GVs in eQTLS (*p*-value = 7.392 · 10^−10^). The colocalization analysis to identify GVs associated with both MD GWAS and cis-eQTLs resulted in 5 GV–eGenes pairs (i.e., genes whose expression is regulated by these GVs; rs10149470—BAG5, rs10149470—RP11-894P9.2 [ENSG00000258851.1], rs12624433—SLC12A5, rs198457—MYRF, rs301799—RP5-1115A15.1 [ENSG00000232912.5]), with a colocalization probability greater than 10% (Table 1). BAG5 and SLC12A5 are involved in neuron projection [44,45] and MYRF in central nervous system myelination [46]. In addition, all eQTLs have been previously associated with MD and other mental disorders according to DISGENET plus [30,47,48] (Appendix A). The eGenes BAG5, SLC12A5, and MYRF show higher expression levels in brain regions according to GTEx (Appendix A). Little is known about the function of the long non-coding RNAs RP11-894P9.2 and RP5-1115A15.1.

### 3.3. MD Associated GVs Affect the TFBS in Regulatory Regions of Genes Relevant for the Disease

The initial set of 5723 GVs associated with MD was first mapped to transcription factor binding sites (TFBS) using Chip-Seq data from ENCODE. A total of 955 GVs were identified as potentially altering the TFBS of 155 TFs (Figure 2). The GVs’ functional impact was assessed with VEP, and 4 sets were created: (a) intergenic and UTR GVs (333), (b) intronic GVs (562), (c) regulatory GVs (303), and (d) non-coding GVs (389). In addition, we further selected those transcription factors (TFs) that were expressed in brain tissues (≥2 TPM), which left 115 TFs.

Using a pattern matching approach (*variation-scan*) [49], we identified GVs likely affecting TFBS. As negative controls, we permuted TF motifs and randomly selected variants matching GVs properties (see Methods). Using permuted motifs and randomly selected variants (1:10) as negative controls, we obtained a total of 306 GVs significantly altering the TFBS of 102 TFs (considering the 4 sets together). Ultimately, 289 GVs and 101 TFs passed the statistical analysis using randomly selected variants (1:1000) as negative control. From this final set, 171 GVs are predicted to disrupt the TFBS of 89 TFs, whereas 143 GVs are predicted to create a TFBS for 82 TFs (Appendix A). Most of these GVs were not characterized as potentially pathogenic by CADD except for 11 GVs (score ≥ 15).

A total of 270 GVs lie in active regulatory regions of the genome of brain-related tissues and cell types according to the epigenome annotation from the ENCODE project based on ChromHMM data (Appendix A) [25,26]. We then looked for evidence of their impact on gene expression regulation by evaluating their annotation to GTEx eQTLs fine-mapped via PICS. The only GV in this dataset of 270 GVs that also fulfills the criteria of being causal and overlapping GWAS and eQTL PICs in the brain with a probability greater than 10% was rs12624433, which is an eQTL for the gene SLC12A5. This GV is predicted to significantly alter the TFBS of USF1, USF2, and MYC. Both rs12624433 and SLC12A5 have been previously associated with major depression disorder and other mental disorders such as bipolar disorder or schizophrenia [48].

In addition, we also inspected the hTFtarget database [27], looking for evidence of a mechanistic association between the eGenes, considered the targets, and the TFs whose binding site is being altered by the GVs. Focusing on brain regions, we have evidence for two GV-TF-eGene/target associations (rs11227217: RAD21 -> ZNRD2-DT [ENSG00000260233.3]; rs62259947: YY1 -> P4HTM). 

The GV rs62259947 has been annotated as an eQTL downregulating the expression of P4HTM in the Brain Cerebellar Hemisphere. We propose this effect is likely being mediated by the GV significantly changing the affinity for YY1 binding (weight difference = 14.98 and *p*-value ratio = 5058.82) (see Methods), a TF known to participate in gene regulation though looping of the DNA [50]. The eGene P4HTM has been associated with the hypoxia-inducible factor HIF1α mediating calcium signaling [51], and its null mutation reduces behavioral despair [52] (Figure 3).

## 4. Discussion

Despite the large volume of genetic information available, the pathogenesis and etiology of MD are not yet fully understood, probably because most GVs lie in non-coding regions with no obvious direct effect on a gene or function. In this context, leveraging multiple omics data is key for moving forward in the understanding of the influence of genomic variants in MD disease development. On top of that, full-genome summary statistics are not readily available due to study sharing policies (especially for private–public research partnerships) hampering the usage of most post-GWAS data analysis tools. This study aims to unravel the role of MD GVs in genetic regulation by focusing on regulatory variation following two complementary approaches: cis-eQTLs and TF binding alterations. Both are key to identifying potentially causal genes and understanding gene expression regulation [6,8], as reported by supporting evidence for its association with other mental disorders [53,54,55] and with MD in particular [56,57,58]. The regulatory variation analysis pipelines we have implemented involve fine-mapping, cis-eQTL colocalization, transcription factor binding analysis, and chromatin accessibility data, specially designed to perform well when full-genome summary statistics are not available [59]. These pipelines are in line with other approaches that leverage available omics data, and as such, they could be applied to other complex disorders with a similar genetic architecture and similar data access issues [53,60,61].

Multiple GVs have been associated with MD, most of them characterized as not potentially pathogenic in addition to being common and mostly in non-coding regions of the genome according to CADD and VEP, respectively (Figure 1). The fine-mapping of MD GVs identified 172 causal GVs and 95 pGenes (Appendix A). The functional enrichment analysis of pGenes stands along with hypotheses of MD pathogenesis such as alterations in neurogenesis and neuroplasticity or the circadian rhythm theory [31]. Additionally, these are also enriched for other phenotypes frequently co-occurring with MD, such as alterations of body mass index or smoking [32]. While most pGenes (63%) and GVs (58%) have previous evidence for association with MD, our study pinpoints novel pGenes and GVs (Appendix A). Additionally, existing literature supports the role of pGenes in processes related to MD pathogenesis, such as immune response, nervous system development, response to stress, or behavioral despair. 

MD causally associated GVs are those most likely to be causal and functioning as eQTLs and, indeed, proved to be significantly enriched in cis-eQTLs from GTEx, in line with previous findings on MD and other psychiatric disorders [53,62]. The colocalizing eGenes are involved in processes relevant to MD, such as neuron projection [63], and have been associated with MD and related phenotypes according to DISGENET plus [47,48]. BAG5 is constitutively expressed in all tissues, while MYRF and SLC12A5 show higher levels in brain tissues (Appendix A). BAG5 has been previously identified as associated with MD [64]. We characterize SLC12A5, involved in chloride homeostasis in neurons, as a pGene, also, and its downregulation has been described as an effect of stress leading to the activation of the hypothalamic–pituitary–adrenal axis, which ultimately can lead to MD-like symptoms [31,65]. However, rs12624433 is an eQTL in the Brain Putamen basal ganglia associated with the upregulation of SLC12A5. Thus, more research is needed to unravel the exact mechanism by which rs12624433 exerts its role in the regulation of the expression of SLC12A5. This eGene has been described as a potential drug target for mental disorders, but considerations should be taken given its important role in brain development; besides, it is highly influenced by exercise and environmental factors [65]. rs198457 mediates the downregulation of MYRF expression, which plays a role in myelination and oligodendrocyte differentiation [46]. These, in turn, require thyroid hormones for their differentiation and maturation [66]. Furthermore, oligodendrocytes have been stated as crucial for psychological functions likely involved in mental disorders such as MD [67].

The analysis of TF regulation with RSAT enabled a precise prediction of TF binding alterations. Although TF expression is not highly tissue-specific [7,68], for this type of analysis, it is key to pick meaningful sets of TFs and GVs [69]. We focused on TF expressed in brain-related tissues as it has been previously reported that genes involved in depression are highly expressed in brain regions [4,32,37,47]. Our analysis resulted in the prediction of 270 GVs lying in active regulatory regions of the genome of brain-related tissues based on chromatin accessibility data. These GVs alter the binding of 101 TFs, roughly equally distributed as disrupting or creating a binding site. The activating or repressing role of these TFs is hard to interpret since it will always depend on the sequence context and the cofactors involved [68]. Thus, further analysis is required to elucidate the impact of these GVs on gene expression regulation. Our pipeline enabled us to filter and prioritize the large number of candidate GVs by combining different omics data and ultimately propose mechanistic hypotheses.

By using eQTL data, we were able to identify the GV rs12624433, which regulates the expression of SLC12A5. This GV, previously referred to as colocalizing, is predicted to alter the binding of the TFs USF1, USF2, and MYC; these belong to the bHLH family involved in neural development [70]. USF1 and USF2 generally exert activating effects [71], with USF1 being a risk gene for Alzheimer’s disease and relevant for brain cholesterol metabolism involving its transport from astrocytes to neurons [72].

Additionally, we found mechanistic evidence for 2 GV-TF-eGene/target associations (rs11227217: RAD21 → ZNRD2-DT; rs62259947: YY1 → P4HTM) when combining pattern matching results, chromatin accessibility data, GTEx eQTLs PICS, and hTFtarget data. Variant rs11227217 is more than 20 kbp away from ZNRD2-DT, but RAD21 is a member of the cohesion complex, which enables genes and enhancers to interact via loop formation [73,74]. NRD2-DT is a lncRNA, and interestingly, our findings include several IncRNAs in the set of pGenes as well as related with regulatory variations, either colocalizing with cis-eQTLS (RP11-894P9.2 and RP5-1115A15.1) or with mechanistic evidence for its association with gene expression regulation (ZNRD2-DT). Although not their exact role in MD pathophysiology is not clear, ncRNAs have been described as promising biomarkers and drug targets for MD [75,76].

Regarding the association rs62259947: YY1 → P4HTM, P4HTM has been related to neurological disorders and social behavior (Figure 3) [51,52]. It is involved in Ca^2+^ signaling mediated by the hypoxia-inducible factor HIF1α altering astrocytes gliotransmission [51]. Indeed, hypoxia has been associated with mental disorders in general and MD in particular [77,78,79,80]. In addition, P4HTM null mutation results in a reduction in fear and depression [52]. In turn, rs62259947 downregulates the expression of P4HTM and changes the binding affinity of YY1 in the Brain Cerebellar Hemisphere. Additionally, YY1 regulates transcription by forming loops, although its specific role as activator or repressor is not yet fully understood [50]. Furthermore, P4HTM and HIF1α have been reported as potential drug targets for MD [52,81]. rs11227217 and RAD21 are associated with red blood cell and reticulocyte count, respectively, by PheWAS [15]. Indeed, red blood cell parameters have been described as altered in patients with mental disorders [82].

## 5. Conclusions

Overall, we have successfully developed and applied a regulatory variation analysis pipeline including fine-mapping, colocalization, TF regulation analysis, and chromatin accessibility data, which overcomes the limitation of the lack of full-genome summary statistics. We have identified causal GVs, pGenes, and eGenes and proposed hypotheses for their role in MD pathogenesis, highlighting the role of chloride homeostasis and myelination. We also found mechanistic evidence involving hypoxia response mediated by altered TF binding. Our findings include GVs and genes supported by the literature on MD and mental disorders, as well as novel molecular mechanisms underlying MD pathogenesis.

## Data Availability

This study analyzed data generated by other projects, which are publicly available as specified in the Methods and Results sections of this paper and summarized here. The GWAS data is available at the GWAS Catalog repository, Accession Study: GCST007342 and PICS Data Portal, https://pics2.ucsf.edu/Downloads/PICS2-GWAScat-2021-10-29.txt.gz (accessed on 1 November 2021) and https://pics2.ucsf.edu/Downloads/GTEx/ (accessed on 2 November 2021). The GTEx RNA-Seq data can be downloaded from https://www.gtexportal.org/home/datasets (accessed on 18 February 2021) (filename: GTEx_Analysis_2017-06-05_v8_RNASeQCv1.1.9_gene_median_tpm.gct.gz); ENCODE data can be accessed from the following Accession Numbers: ENCSR674KAN, ENCSR801APH, ENCSR826BFW, ENCSR658SFK, ENCSR082KYZ, ENCSR363VGK, ENCSR738WFF and ENCSR860PXK. The data from JASPAR, ENCODE, HOCOMOCO, footprintDB and hPDI is available at (http://rsat.sb-roscoff.fr/retrieve-matrix_form.cgi (accessed on 24 February, 2 and 4 March 2021), see View matrix descriptions and download full collections); and hTFtarget data can be downloaded from http://bioinfo.life.hust.edu.cn/hTFtarget#!/download (accessed on 15 November 2021) (filename: TF-Target-information.txt). The data generated by the current study as a result of the analysis of the above-mentioned datasets are available at Zenodo (https://doi.org/10.5281/zenodo.6838470) and are also available in the Appendix A with this manuscript.

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
