# Peer review of "Functional Genomics Analysis to Disentangle the Role of Genetic Variants in Major Depression"

_genes, 2022, doi:10.3390/genes13071259_

Round 1
Reviewer 1 Report
Papers can be recommended for minor revisions and other reviewers/editors suggestions must be considered.
Author Response
The authors want to thank the reviewers for dedicating time to evaluate our submission and for providing positive and constructive feedback. We made our best to address the reviewers’ comments.
The revised version of the manuscript includes changes introduced to address the reviewers' comments and suggestions, including references to other studies in the discussion section.
Reviewer 2 Report
In this manuscript, the authors have examined the available GWAS data for Major depression using bioinformatic tools to determine the cis-eQTLs colocalization and analysis of transcription factor binding sites for the genetic variants. This analysis identified novel related genes, pathways and transcription factors which can affect the pathogenesis of major depression and can provide further mechanistic understanding of a complicated disorder.
The manuscript is well written and the analysis if the data has been clearly explained.
1) Can the authors give more details about the study by Howard et al, so as to have a clear picture of why this data is significant in terms of the analysis being performed. Why only this dataset was selected for this study?
2) Please provide a succinct schematic if the process of the analysis using the full forms of the terms like CADD, VEP etc? It can be e.g. a combination of figure 1 and scheme 1. A simple tabular format will suffice in order to get the big picture of the steps involved in the analysis which have been described in the Materials and methods section.
3) The effect of Rad21 which is also an architectural protein like YY1 can be described in the discussion along with YY1 and some commonalities can be pointed out as both these proteins are involved in looping.
4) The authors can also probably extend their methodology to other disorders and speculate if the method can be used in case of other disorders.
Author Response
The authors want to thank the reviewers for dedicating time to evaluate our submission and for providing positive and constructive feedback. We made our best to address the reviewers’ comments. We provide a point-by-point answer to their comments and introduced changes in the manuscript files to address the issues raised by them.
1) Can the authors give more details about the study by Howard et al, so as to have a clear picture of why this data is significant in terms of the analysis being performed. Why only this dataset was selected for this study?
We selected this study because it was the most recent GWAS on MD with accessible summary statistics when this project started. The study by Howard et al is a meta-analysis of the three largest GWAS studies on MD at that time[1],[2],[3], with 807,553 individuals, including 246,363 cases and 561,190 controls. Thus, in our opinion, it was the most suitable dataset for performing the post-GWAS data analysis that we propose in our manuscript.
2) Please provide a succinct schematic if the process of the analysis using the full forms of the terms like CADD, VEP etc? It can be e.g. a combination of figure 1 and scheme 1. A simple tabular format will suffice in order to get the big picture of the steps involved in the analysis which have been described in the Materials and methods section.
We have incorporated a new Table (Supplementary Table S1) in the Supplementary Tables file containing all the resources used for the analysis pipeline and their purpose. Additionally, in the manuscript, schema I depicts the fine-mapping and the colocalization analysis while schema II describes the transcription factor binding site analysis. In addition, the Supplementary schema I presents an overview of the whole study. We have also included in the manuscript an Abbreviations section and a full description of abbreviations used in the Figures, Schemas, Tables and Supplementary Material footnotes. We hope these changes make easier the reading of the manuscript.
3) The effect of Rad21 which is also an architectural protein like YY1 can be described in the discussion along with YY1 and some commonalities can be pointed out as both these proteins are involved in looping.
We thank the reviewer for this suggestion. We have incorporated this idea with supporting references in the discussion section of our manuscript (page 12) as follows:
“Besides, we found mechanistic evidence for 2 GV-TF-eGene/target associations (rs11227217: RAD21 → ZNRD2-DT; rs62259947: YY1 → P4HTM) when combining pattern matching results, chromatin accessibility data, GTEx eQTLs PICS and hTFtarget data. Variant rs11227217 is more than 20kbp away from ZNRD2-DT, but RAD21 is a member of the cohesin complex, which enables genes and enhancers to interact via loop formation [74, 75]. ZNRD2-DT is a lncRNA and interestingly our findings include several IncRNAs in the set of pGenes as well as related with regulatory variations, either colocalizing with cis-eQTLS (RP11-894P9.2 and RP5-1115A15.1) or with mechanistic evidence for its association with gene expression regulation (ZNRD2-DT). Although not being clear their exact role in MD pathophysiology, ncRNAs have been described as promising biomarkers and drug targets for MD [59, 60].”
4) The authors can also probably extend their methodology to other disorders and speculate if the method can be used in case of other disorders.
We thank the reviewer for this suggestion. We have incorporated a comment in the Discussion (page 11) including some references for studies that apply similar strategies for complex diseases. The paragraph reads as follows: “These pipelines are in line with other approaches that leverage available omics data and as such, they could be applied to other complex disorders with a similar genetic architecture and similar data access issues [53, 60, 61].”
[1] Howard, David M., et al. "Genome-wide association study of depression phenotypes in UK Biobank identifies variants in excitatory synaptic pathways." Nature communications 9.1 (2018): 1-10.
[2] Hyde, Craig L., et al. "Identification of 15 genetic loci associated with risk of major depression in individuals of European descent." Nature genetics 48.9 (2016): 1031-1036.
[3] Wray, Naomi R., et al. "Genome-wide association analyses identify 44 risk variants and refine the genetic architecture of major depression." Nature genetics 50.5 (2018): 668-681.
Reviewer 3 Report
This study performed a fine-mapping analysis of genetic variants associated with major depression and applied a pipeline focused on gene expression regulation through two complementary approaches to identify putative causal variants with proximal genes associated with the pathophysiology of major depression. This study can provide some theoretical support for the study of brain-related diseases. It has a good theoretical value.
Author Response
The authors want to thank the reviewers for dedicating time to evaluate our submission and for providing positive and constructive feedback. We made our best to address the reviewers’ comments. We provide a point-by-point answer to their comments and introduced changes in the manuscript files to address the issues raised by them.